# Pharmacogenetics to Avoid Loss of Hearing (PALOH) trial: a protocol for a prospective observational implementation trial

John Henry McDermott [1,2] Rachel Mahood,[1] Duncan Stoddard,[1,3] Ajit Mahaveer,[4] Mark A Turner,[5] Rachel Corry,[1] Julia Garlick,[1] Gino Miele,[6] Shaun Ainsworth,[6] Laura Kemp,[6] Iain Bruce,[7] Richard Body,[8,9] Fiona Ulph,[10] Rhona Macleod,[1] Karen Harvey,[5] Nicola Booth,[4] Peter Roberts,[11] Paul Wilson [12] William G Newman[1,2]

## ABSTRACT

**Introduction** In conjunction with a beta-lactam, aminoglycosides are the first-choice antibiotic for empirical treatment of sepsis in the neonatal period. The m.1555A>G variant predisposes to ototoxicity after aminoglycoside administration and has a prevalence of 1 in 500. Current genetic testing can take over 24 hours, an unacceptable delay in the acute setting. This prospective-observational trial will implement a rapid point of care test (POCT), facilitating tailored antibiotic prescribing to avoid hearing loss.

**Methods and analysis** The genedrive POCT can detect the m.1555A>G variant in 26 min from buccal swab. This system will be integrated into the clinical pathways at two large UK neonatal centres over a minimum 6-month period. The primary outcome is the number of neonates successfully tested for the variant out of all babies prescribed antibiotics. As a secondary outcome, clinical timings will be compared with data collected prior to implementation, measuring the impact on routine practice.

**Ethics and dissemination** Approval for the trial was granted by the Research Ethics Committee (REC) and Human Research Authority in August 2019. Results will be published in full on completion of the study.

**Trial registration number** ISRCTN13704894.

**Protocol version** V 1.3.

For numbered affiliations see end of article.

**Correspondence to**
Dr John Henry McDermott;
john.mcdermott2@mft.nhs.uk

### Strengths and limitations of this study

► This manuscript describes a prospective observational implementation trial of a rapid genetic point of care test (POCT) in the acute neonatal setting to tailor antibiotic therapy, aiming to reduce aminoglycoside-induced hearing loss.

► We embed a genetic POCT within the admission pathways of two neonatal intensive care units and, due to time pressure, prospective consenting is not possible, requiring the implementation of a unique ethical framework described within the protocol.

► To our knowledge this is the first example of a genetic POCT implemented in the acute neonatal setting to alter management.

► Rather than assessing the characteristics or performance of the assay itself during the trial, the primary and secondary outcomes focused on the utilisation of the system and whether normal clinical practice was impacted following implementation.

► A limitation is that, given the relatively rare nature of the variant (1 in 500), it may transpire that the variant is never detected throughout the study, meaning some secondary outcomes cannot be reliably assessed.

## BACKGROUND

Aminoglycosides are broad-spectrum antibiotics which act by binding to the 16S rRNA component of the bacterial 30S ribosomal subunit, resulting in the translation of truncated proteins.[1 2] These abnormal proteins stimulate a stress response within the bacteria culminating in cell death. Due to their low cost and effectiveness, they are one of the most frequently prescribed medicines globally.[3] The National Institute for Health and Care Excellence (NICE) advises the use of intravenous benzylpenicillin with gentamicin as the first-choice antibiotic regimen for empirical treatment of infection in the neonatal period.[4] This combination has the major advantage of having a narrow spectrum of activity and lower risk of antibiotic resistance compared with alternative antibiotic regimens, such as third generation cephalosporins which are recommended as second line agents.

The side effect profile from protracted courses of aminoglycosides is well known, with nephrotoxicity and ototoxicity commonly recognised in cohorts who receive large amounts of aminoglycoside.[5] An ototoxic

compound is one that can cause damage to the auditory system (cochleotoxicity) or, less commonly, vestibular system (vestibulotoxicity). Careful analysis revealed that aminoglycoside-induced ototoxicity (AIO) clusters within families and this trait seemed to be transmitted maternally, suggesting mitochondrial inheritance. Further work demonstrated that this susceptibility was caused by a variant in the 12S rRNA (RNR1), m.1555A>G.[6] This causes a change in the conformation of the 12S rRNA, producing a structure more like the bacterial 16S rRNA, meaning aminoglycosides can more readily bind causing cellular toxicity. The impact of this aberrant binding is most apparent in the inner ear hair cells, leading to ototoxicity.

The m.1555A>G variant is strongly associated with AIO and has a reported prevalence of 0.2% (~1 in 500).[7] It has previously been suggested that genetic testing should be used in children requiring aminoglycosides to prevent hearing loss and that this approach would be cost-effective when balanced against the costs of lifelong deafness.[3 7] In many centres, children with cystic fibrosis are tested for the variant at diagnosis as it is expected that these individuals will require aminoglycoside antibiotics at some stage in their lives. This is typically undertaken by a validated genotyping assay in an accredited diagnostic laboratory, a strategy which takes approximately 3–4 days to return a result. Understandably, this testing strategy is not suitable for use in clinical settings where administration of antibiotics is required within an hour of the decision to start treatment with antibiotics.

Currently, there is an inability to test for the m.1555A>G variant in the acute setting such as babies admitted to the neonatal intensive care unit (NICU), representing approximately 90 000 admissions per year in the UK alone. By introducing a simple genetic point of care test (POCT) with a result in under 30 min, we have the potential to avoid approximately 180 cases of severe/profound irreversible deafness every year in the UK alone. Those babies with the m.1555A>G variant could be prescribed an equally effective second line antibiotic, such as a cephalosporin. The UK NICE guidance for early onset neonatal infection recommends a combination of gentamicin and benzylpenicillin as first-line therapy not due to superiority over other agents on an individual patient level, but because that regimen has a narrower spectrum of activity and therefore does not readily contribute towards the development of resistant bacterial pathogens.[4]

Alongside an industry partner, genedrive, we have a developed a small and robust thermocycler platform for rapid point-of-care diagnostic testing of the m.1555A>G variant which has been CE certified for clinical use. This protocol describes a multi-centre prospective observational trial to assess the implementation of the POCT in the clinical setting.

## METHODS
### Study design
Investigator-initiated, multi-centre, prospective-observational, implementation trial. Two study sites, both of which are recruiting and experimental.

### Primary objective
To critically assess the performance of a newly developed genetic POCT for use on the NICU and measure whether the clinical teams can integrate the test into their clinical practice without disrupting normal standard of care, facilitating tailored antibiotic prescribing.

### Secondary objectives
Exploratory objectives include an assessment of the wider impact of the implementation, including impact on resources and clinical timelines. We will also assess instances where the assay was not used and consider reasons for deviation. The reliability of the assay itself will also be measured through regular monitoring of results, assessment of test failures and confirmation against gold-standard genotyping.

### Study centres and time schedule
Participants will be recruited from two large UK based NICUs in the UK, Manchester University National Health Service (NHS) Foundation Trust and Liverpool Women's NHS Foundation Trust (LWH). Both sites will recruit participants and conduct the trial. It should be noted that there is considerable variation in antimicrobial prescribing practices both between and within different countries. The type of antibiotics used, dosing regimens and monitoring protocols can vary between different nations, cities and institutions.[8] Both units in this study use a combination of a beta-lactam and an aminoglycoside as first line therapy for early onset neonatal infection. Based on historical data, the majority of admissions to both units will be screened for infection and, as such, considered for antibiotic therapy. The trial began in January 2020 and the last participant was recruited in November 2020. The study continued despite the SARS-CoV-2 pandemic.

### Primary outcome
The primary outcome is the number of neonates who are successfully tested for the m.1555A>G variant via this novel POCT genetic test as a proportion of all babies who receive aminoglycoside antibiotics on the two participating NICUs.

### Secondary outcomes
This trial will also assess several secondary outcomes including the total number of neonates identified with the m.1555A>G genetic variant (positive result) as a proportion of the study population. An assessment of resource impact will be undertaken, including a health economic analysis of additional staff time to secure sample and testing and costs associated with alternative antibiotic prescribing for any positive tests. We will record the number of neonates where testing was not

undertaken and the rationale for this. Correlation of the POCT result with the current in-house reference assay will be assessed and any test fails will be reported. In the acute neonatal setting, delivering antibiotics in a timely manner is critical and this should ideally be within an hour after admission. As such, we will assess whether the time to antibiotic administration after admission is impacted by implementation of the genetic POCT. A reference time to antibiotic was measured over a month-long period before implementation.

### Inclusion criteria

All babies admitted to NICU across the two participating sites, commencing from the trial start date. This includes babies admitted directly from delivery suite, midwifery led units or transferred from another neonatal unit. At the LWH site, babies who are screened for infection on the NICU but then transferred to external wards (not formally admitted to NICU) will also be included. It is a clinical decision whether the babies are being screened for infection as part of their assessment, reflecting the real-world, pragmatic, nature of the trial. This also reflects the variation in admission procedures between the two trial sites where pathways differ due to local practices.

Given that all babies admitted to NICU are eligible for recruitment, some of those recruited will not go onto receive antibiotic therapy. If the baby did not go onto receive antibiotics, their data will still form part of the dataset for analysis.

### Exclusion criteria

Babies requiring antibiotics immediately, as determined by the admitting clinician, with already established intravenous access. These exclusion criteria recognise the potential urgency of any decision to deliver antibiotics. If intravenous access has not been achieved, then this may provide time to run the assay. We expect that only a small proportion of admissions would be excluded based on these criteria.

### Training

Training of a minimum of 80% of relevant nursing and medical staff within the two NICUs will commence in the 6 months prior to study start. This figure has been chosen, based on guidance from senior NICU clinicians and nursing staff, to ensure that there will always be staff members on-shift who are trained to undertake the assay throughout the study.

Training will include practical use and interpretation of the assay, with Standard Operating Procedures for use integrated into the standard admission procedure. A 'train the trainer' approach will be adopted, where a number of experienced NICU research nurses plus additional clinical nursing staff identified as 'super-users' will receive training directly from representatives of the device manufacturer. These designated super-users will then cascade training to the remaining nursing and medical staff within the two neonatal services.

Ongoing refresher training and online resources will be made available throughout the trial. A training log will be maintained detailing staff competency including dates of training, in order to identify any correlation of assay performance with increasing training numbers. NICU research nurses and coinvestigators will be fully Good Clinical Practice (GCP) competent. A local, named coinvestigator (neonatal consultant) will be responsible for clinical staff performing any duties in accordance with the protocol, GCP and local requirements.

### Testing process and assessment

Participants will be recruited as they are admitted or reviewed by the NICU team. Given the acute nature of the project and the timeframes involved, there is no opportunity to undertake a formal pretrial screening assessment or prospective consenting process. All babies admitted to NICU or screened for infection by the NICU team will be deemed eligible for recruitment unless there is a clinical decision by the responsible admitting clinician that the exclusion criterion is met and the individual requires immediate antibiotic therapy, without delay.

Following recruitment, the time that a decision was made to prescribe antibiotics represents time 0 (figure 1). If antibiotics were not prescribed, then time 0 is represented by the time of admission. The genetic POCT is performed from a buccal swab. When babies are admitted to NICU several swabs are already taken at admission. Where babies are recruited to the study, the m.1555A>G buccal swab will be performed alongside these normal admission processes. The assay will be commenced, and normal practice will continue. Once the assay result is available, it can be used to then undertake personalised antibiotic prescribing, avoiding aminoglycoside antibiotics if the m.1555A>G variant is detected. The CE certified assay will be integrated into the admission pathways at both centres and clinicians will not be blinded to the results.

As part of the study the following variables will be recorded; participant demographics, admission time, time of decision to prescribe, time swab taken, time assay started, time of assay result, assay result, time of antibiotic prescription, antibiotics prescribed. Where a positive result is identified and antibiotic therapy is required, an alternative antibiotic regimen (standard non-gentamicin pathway as per hospital guidelines) will be prescribed and a referral should be made to clinical genetics for familial cascade testing. Where there is a test-fail, standard of care should be followed, that is, the baby should be administered gentamicin if clinically indicated. Negative results will not be formally reported to the parents. However, all parents will be provided with information on the study in their information pack, allowing them to contact the study team for further information if they wish.

Assay performance will be assessed throughout the study, with at least 10% of samples each month tested via gold-standard sequencing. Any positive samples will also be confirmed as per standard clinical pathways.

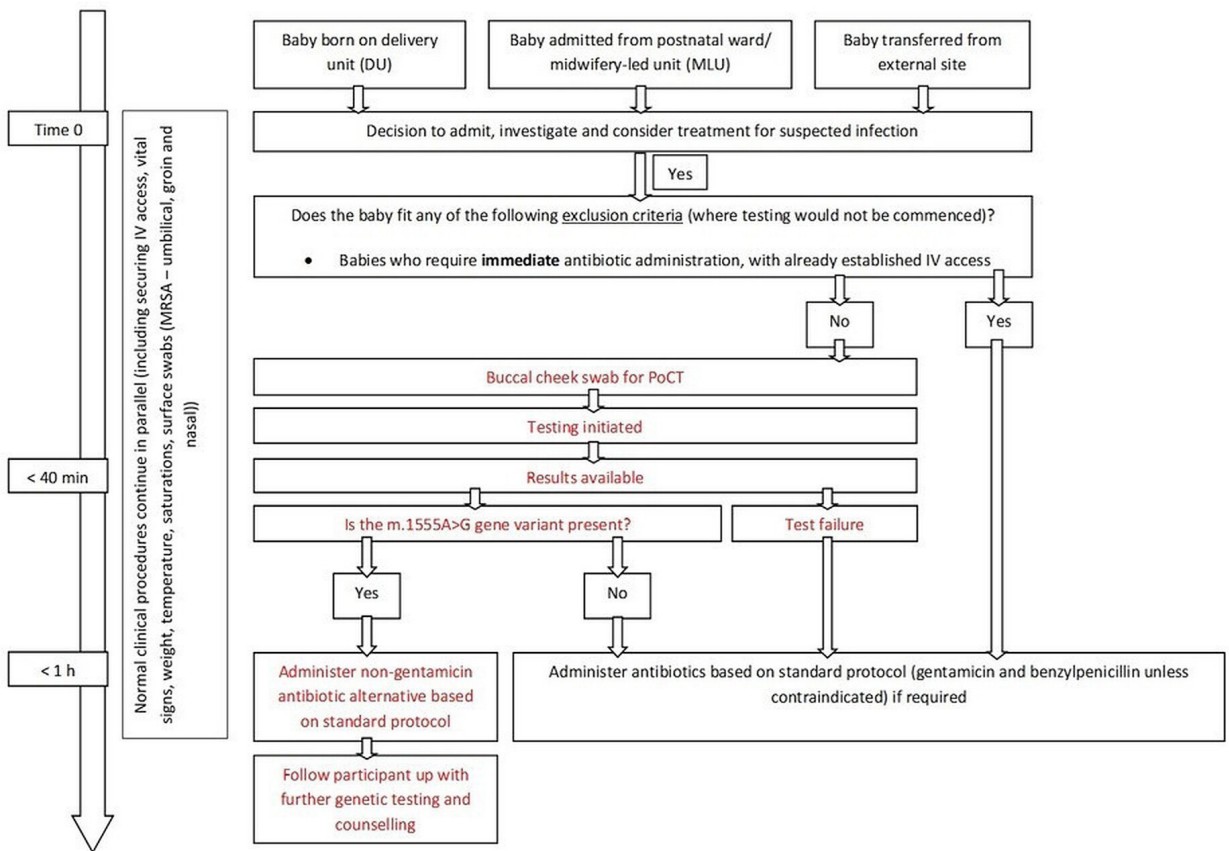

**Figure 1** Flow diagram demonstrating the admission, testing and prescribing process in the Pharmacogenetics to Avoid Loss of Hearing trial. The time that a decision was made to prescribe antibiotics represents time 0. If antibiotics were not prescribed, then time 0 is represented by the time of admission. POCT, point of care test.

Any divergence between the POCT and gold-standard methodology will be reviewed by the trial management committee.

Data collection will be undertaken contemporaneously by the clinical staff and any missing data will be collected by the NICU research nurses. Data will be entered into an electronic database by the clinical team and a study ID will be generated. The participants will subsequently be anonymised to the research team. If an individual is not tested for any reason (eg, test overlooked, staff availability or the individual meets the exclusion criteria), these incidences will be captured retrospectively by NICU research nurses by checking daily admission records and the reasons will be documented.

### Samples size

The sample size (approximately 900 babies) is estimated based on the expected number of admissions across both clinical sites over an approximate 10-month trial period. To achieve a statistical power of 90% with a 5% significance level, a margin of 0.5 (in Cohen's d, which equates to about 7 min difference given the SD in the control sample) and a control sample of N=50, a test sample size of 112 is required. With the forecast test sample size of 900, statistical power should be around 95% to detect variance in clinical timings before and after implementation.

### Statistical analysis

For the primary outcome, simple descriptive statistics will be used to provide a metric for the proportion of babies tested for the variant over the number who were ultimately prescribed aminoglycoside antibiotics.

Regarding the secondary outcome measures, the objective is to establish that the introduction of the POCT does not significantly impact the time between admission to NICU and the prescription of antibiotics for acute admissions. For the purposes of this analysis, those babies receiving antibiotics within the first 2 hours of admission, representing acute antibiotic administration, will be compared. The burden of proof rests on the research hypothesis, meaning the null hypothesis is that the new POCT does impact this quantity. This is a so-called non-inferiority significance test. The quantity of interest is the mean difference between the admission to ICU and prescription of antibiotics timestamps in two situations: with and without the POCT test ($\mu_c, \mu_t$). The null hypothesis is that the difference between these quantities is greater than or equal to some prespecified margin ($\Delta$) that is, $H_0 : \mu_c - \mu_t \geq \Delta; H_A : U_t - U_c < \Delta$. A type I error here is concluding the time durations are equivalent when in fact the time to antibiotic with the POCT is greater.

The two samples are of unequal size, with the control ('before implementation') sample having about $n_c$=50,

and a forecasted test sample size of around $n_t=900$. We also have no insight into population variance and cannot assume they are the same in both groups (it may eg, take some staff members longer to carry out the new test than others). As such, a one-sided Welch t-test will be used to reject or not reject the null hypothesis.

Throughout the study we anticipate that there will be iterative updates to the POCT technology, as avenues for optimisation emerge because of our findings from real-world implementation. The timings of these updates will be recorded and separate post-hoc statistical analysis will be undertaken to assess the impact of each update.

## Study organisation

Two groups have been convened to oversee the trial, a management group and a steering committee. The Trial Management Group convenes monthly to ensure all practical details of the trial are progressing well, within the agreed milestones, and is led by the study chief investigator. The trial management group will act as the data monitoring committee. Membership is independent of the sponsors. The data management team will have access to anonymised data and will review assay performance and trial conduct on a monthly basis-specifically the concordance with gold standard. All data will be analysed by an independent statistical team who report monthly to the trial management group. Any amendments to the protocol will be communicated to the REC, funding agency and the trials registry.

The Trial Steering Committee meets on a 6 monthly basis to monitor the performance of the trial against the agreed project plan and advise on scientific and technical aspects of the project. The membership is independent of the sponsor and investigators.

A stakeholders group has also been assembled to advise on the design and dissemination of the study intervention in the interests of the intended end users and participants. This membership includes personnel from key clinical services (neonatology and paediatric audiology), commissioners (NHS England) and newborn and hearing loss charities.

## Patient and public involvement

Involvement of parent and public representatives has been a critical component of the development of this trial protocol. Parent representatives are involved in both the trial management groups, stakeholders committee and a separate Public and Patient Involvement and Engagement (PPIE) panel, providing both neonatal care and/or hearing loss experience. The PPIE panel was involved in the development of both the protocol and ethics application. The PI and coinvestigators presented early versions of the texts to the PPIE panel and any potentially contentious issues were discussed in more detail. The insights gained from these meetings were used to refine the applications prior to submission.

## Ethics and adverse events

This is the world's first trial, that we are aware of, of a genetic POCT to personalise medicine in the acute neonatal setting. As such, there have been several ethical challenges, mainly performing DNA analysis without qualifying consent. Prospective consent is not possible given the acute nature of the admission, therefore we sought permission to proceed with the POCT included as part of the standard of care. The parents could then opt-out of the data being used as part of the study. This ethical process has been described in detail previously.[9] Approval was provided by the Research Ethics Committee (REC) and the Human Research Authority (HRA) in August 2019. We will record any adverse event related to the sample collection procedure or directly related to the testing process.

## Dissemination of results

The results of the study will be submitted to international peer-reviewed scientific journals, irrespective of their outcome and presented as per best practice Strengthening the Reporting of Observational Studies in Epidemiology guidelines for observational trials.

## DISCUSSION

To our knowledge, the PALOH study represents the first trial of a genetic POCT designed to alter management in an acute, time-sensitive setting. Successful implementation could avoid up to 180 cases of aminoglycoside-induced ototoxicity in the UK each year, and many more worldwide. However, this study has implications beyond this gene–drug pair. The anticipated results will indicate whether genetic technology can be implemented in the acute setting more broadly, a hitherto untested concept. The unprecedented nature of this work led to several practical and ethical challenges in the design of the protocol.

Several studies have previously reported that most clinicians are not confident with the interpretation and implementation of genetic data.[10–12] With this in mind, we ensured that the assay implemented as part of this trial was as straightforward to interpret as possible, providing a binary readout for the presence or absence of the m.1555A>G variant. However, even if an assay is simple to operate and interpret, we recognised that the imposition of any alien process in such a critical time-period could impact adoption. We therefore designed this protocol to include a comprehensive training strategy prior to the trial commencing on both units. This significant undertaking ensures that over 500 healthcare professionals, greater than 80% of all staff, have exposure to the system before implementation.

Due to CE certification requirements for medical devices in the European Union, any new system must be validated prior to clinical implementation. For a technology which has been designed for use in the acute setting, this requirement poses unique challenges. Although we were

able to thoroughly assess the technology in the preclinical laboratory setting, satisfying the requirements for CE certification, it was not possible to perform analytical validation in its intended use setting, the NICU. This was because any result generated in this setting would be actionable and could impact management, precipitating an ethical dilemma. As a result, we determined that the most appropriate strategy was to regularly review the performance of the assay following implementation and introduce any updates to the training or technology as required. We have therefore made clear in our analysis plan that outcome statistics will be reported for the study as a whole and for each time-period post-update, if any such updates are necessary.

Given the acute context in which the Pharmacogenetics to Avoid Loss of Hearing (PALOH) trial is taking place, it was recognised that it would not be practicable for informed consent to be gained prospectively. Newborns admitted to NICU are, by definition, acutely unwell and it is likely to represent one of the most stressful time in a parent's life. As such, it was determined that asking for consent to perform this genetic test immediately after birth would be inappropriate. Following discussion with stakeholders including parents, neonatologists and geneticists, ethical approval was sought based on the POCT being included as part of the standard package of care offered to the acutely unwell neonate, for which broad parental consent is typically provided, or if unavailable, is undertaken in the best interest of the child.[13] Our PPIE panel were particularly helpful during this process, providing a forum to discuss the various consenting options where we could consider relative benefits and costs of each strategy.

It should be noted that there is considerable variation in antimicrobial prescribing practices both between and within different countries. The type of antibiotics used, dosing regimens and monitoring protocols can vary between different nations, cities, and institutions. The reasons for this are multifactorial, associated with historical practice, the adoption of clinical guidelines and economic considerations.[14–16] Where there are national guidelines, such as those produced by NICE, one would expect relative consistency in prescribing practice. However, where practice is less guideline driven, there may be more variation. This is exemplified by a retrospective cohort study of 52 061 infants from 127 NICUs across California, which found that antibiotic use varied between centres by up to 40-fold.[15] Variability in prescribing practice in the UK is likely to be less pronounced, given the more guideline driven, nationalised nature of its healthcare system. As such, health economic cases for the utilisation of POCT systems, such as the one described here, will vary depending on the wider healthcare context and should be formally examined locally.

Neonates are tested for the variant as part of their admission process via a buccal swab, the variant details then used to inform prescribing and data is subsequently gathered and analysed. Parents will be provided information regarding the trial by the NICU team as part of an existing parental information pack. They would have the right to ask for their child's data to be removed from the study at that stage. This provides a mechanism to opt-out prior to the analysis stage although, critically, by this point their child will have already had tailored antibiotic prescribing based on the presence or absence of the variant.

To our knowledge this represents the first example of such a consenting model being used in a trial involving genetic testing. In the summer of 2019, an application for ethical approval was submitted to the Health Research Authority Research Ethics Service (RES), proposing that the importance of this genetic test and the potential benefits warranted approval of an opt-out consenting model. After consideration, the REC approved the PALOH trial design in full pending HRA approval, making it the first trial of a genetic POCT to alter management in the acute setting. However, in the week following the REC meeting the PALOH study team received notice that the HRA were withholding approval as they had concerns that the study was in violation of the Human Tissue Act (2004) (HT Act). Specifically, the HRA felt that the trial design did not meet the legal definition of 'qualifying consent' as outlined in the HT Act.

Following protracted ethical and legal discussion, the trial design was approved following the agreement that the act of performing the initial test represents a clinical decision, rather than a research question. This approach was then permissible under Schedule 4 Part 2 of the HT Act which outlines that 'the medical diagnosis or treatment of the person whose body manufactured the DNA' is an excepted purpose for DNA analysis without formal consent.[9] We believe that the PALOH study represents the first time this ethicolegal issue has been considered. As such, our methodology may be of value to future clinical academics exploring the application of similar pharmacogenetic biomarkers.

The anticipated results of the PALOH trial will assess whether a genetic POCT can be used to tailor antibiotic prescribing in the acute NICU setting. This will be of value not only in avoiding AIO but also in optimising management in other disease areas, as the relevance of genetics in everyday clinical practice continues to develop. We believe that this trial methodology and the ethical issues outlined in this work will be of value when researchers are designing similar studies in the future.

## Trial status

The PALOH study has closed to recruitment and analysis is ongoing. Recruitment began in January 2020 and completed in November 2020.

**Author affiliations**
[1]Manchester Centre for Genomic Medicine, Manchester University NHS Foundation Trust, Manchester, UK
[2]Division of Evolution and Genomic Sciences, The University of Manchester, Manchester, UK
[3]DS Analytics and Machine Learning Ltd, London, UK

[4]Neonatal Intensive Care Unit, Manchester University NHS Foundation Trust, Manchester, UK

[5]Neonatal Intensive Care Unit, Liverpool Women's Hospital NHS Foundation Trust, Liverpool, UK

[6]Genedrive plc, Manchester, UK

[7]Paediatric ENT Department, Manchester University Hospitals NHS Foundation Trust, Manchester, UK

[8]Emergency Department, Manchester University NHS Foundation Trust, Manchester, UK

[9]Division of Cardiovascular Sciences, The University of Manchester, Manchester, UK

[10]Division of Psychology & Mental Health, University of Manchester, Manchester, UK

[11]Market Access & Reimbursement Solutions, Manchester, UK

[12]Alliance Manchester Business School, University of Manchester, Manchester, UK

**Acknowledgements** The authors would like to thank the patients and family members who took part in the study. We would like to acknowledge the huge efforts of the nurses, physicians and administrative staff at both study sites who have been instrumental in developing the project. We are very grateful to the parents and public representatives who sat on the PPIE committee and advised on the development of the project.

**Contributors** JHM and WGN developed the concept for the study. SA, LK and GM designed and optimised the assay for implementation. All authors contributed to the development of the protocol led by AM and MAT and the Manchester Foundation Trust and Liverpool Women's Hospital sites, respectively. KH and NB manage the data collection at each clinical site. RMahood is research manager for PALOH. DS, PR and PW are involved in data analysis. RMacleod and FU have designed and undertaken stakeholder interviews which contribute to the design of the protocol. JG and RC, along with RMahood, have overseen the PPIE process. RB, IB and PW have advised on the implementation process and broader study design. All authors are responsible for either the recruitment of participants, data collection or data analysis. All authors read and approved the final version of this manuscript.

**Funding** Funding for the project was provided via an NIHR i4i programme award (Grant Number: II-LB-0417-20002), with additional support from the Manchester NIHR Biomedical Research Centre, Hearing Health Theme (IS-BRC-1215-20007). Pilot data support from Action on Hearing Loss Flexi grant F67_Newm.

**Competing interests** SA, LK and GM are employees of genedrive.

**Patient and public involvement** Patients and/or the public were involved in the design, or conduct, or reporting, or dissemination plans of this research. Refer to the Methods section for further details.

**Patient consent for publication** Not required.

**Provenance and peer review** Not commissioned; externally peer reviewed.

**ORCID iDs**
John Henry McDermott http://orcid.org/0000-0002-5220-8837
Paul Wilson http://orcid.org/0000-0002-2657-5780

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
