## [Reviewer comments · BMJ Open]

ARTICLE DETAILS

TITLE (PROVISIONAL)	The Pharmacogenetics to Avoid Loss of Hearing (PALOH) Trial: A Protocol for a Prospective Observational Implementation Trial
AUTHORS	McDermott, John; Mahood, Rachel; Stoddard, Duncan; Mahaveer, Ajit; Turner, Mark; Corry, Rachel; Garlick, Julia; Miele, Gino; Ainsworth, Shaun; Kemp, Laura; Bruce, Iain; Body, Richard; Ulph, Fiona; Macleod, Rhona; Harvey, Karen; Booth, Nicola; Roberts, Peter; Wilson, Paul; Newman, William

VERSION 1 – REVIEW

REVIEWER	Rybak, Leonard P. Southern Illinois University School of Medicine, Otolaryngology
REVIEW RETURNED	24-Nov-2020

GENERAL COMMENTS	This protocol addresses a highly significant problem of genetic susceptibility of neonates to aminoglycoside ototoxicity. It provides an excellent rationale for the study to be done and provides statistical validation of the data to be acquired. The results could later be applied worldwide to prevent hearing loss in neonates with genetic susceptibility to aminoglycoside ototoxicity by providing rapid screening using a novel tool a genetic point of care test. Such a tool could prevent lifelong hearing loss in neonates and promote a substantial improvement in quality of life and reduce health care costs associated with lifelong deafness in infants if alternative antibiotic therapy can be substituted in a timely manner.
---

REVIEWER	Peng, Zhiyu Beijing Genomics Institute Shenzhen
REVIEW RETURNED	20-Dec-2020

GENERAL COMMENTS	This protocol reported a prospective observational trial to assess the implementation of the genetic POCT in NICU. The authors state that this is the first attempt to implement a genetic POCT in the acute neonatal setting. The authors state that when the POCT result is positive, a standard non-gentamicin pathway should be prescribed if antibiotic therapy is required, without detailed description of the non-gentamicin prescription. We doubt this straight-forward determination of antibiotic selection for the infant by genetic POCT, considering that gentamicin is the first-choice antibiotic for neonatal sepsis and it is not clear whether other antibiotic regimens are equivalent to gentamicin in neonatal sepsis treatment. Moreover, there is a conflict between the use of gentamicin that may cause deafness and non-gentamicin
--

	prescription that may increase the risk of death, and an opt-out consenting model might be inappropriate. Additionally, this study plans to recruit ~900 babies. Calculating the prevalence of m.1555A>G (1 in 500), it is possible to never detect carriers in this study. Finally, the manuscript mentioned: "The trial will start in January 2020 and it is expected that the last participant will be recruited in November 2020". However, protocol papers should report planned or ongoing studies. I am confused that is it a completed study or an ongoing study?
--	---

REVIEWER	Garinis, Angela Oregon Health & Science University
REVIEW RETURNED	19-Jan-2021

GENERAL COMMENTS	This is a well-written study protocol by a highly esteemed group of scientists. The clinical implementation of a genetic test to detect mutations associated with ototoxicity is important, and may have considerable clinical impact, particularly in the prevention of aminoglycoside-induced ototoxicity. Our current clinical models are moving to point-of-care programs, which tend to be more efficient and cost-effective. This trial is directly in-line with modern practice. I have reviewed the manuscript in detail and have provided some comments, as well as requests for clarification: 1] Page 5, Line 19: Provide proper citation and reference for NICE guideline (CG149). 2] Page 5, Line 31: Add hyphen "aminoglycoside-induced ototoxicity" 3] Page 5, Lines 38-40: Elaborate on comparison between structures of 12S rRNA and 16S rRNA. This statement was made without clear context. 4] Introduction (general comments): Given the audience of this journal, it would be beneficial to the reader to provide a brief description of both ototoxicity and vestibulotoxicity. 5] Page 6, Lines 17-20: If β-lactam antibiotics are equally effective as aminoglycosides, and produce less adverse events, then why do we continue to use the aminoglycoside? This is an important point of discussion, and is highly relevant to numerous clinical cohorts routinely treated with aminoglycosides. 6] Page 6, Lines 23-30: The authors indicated that the rapid test is comparable to the gold-standard and has been approved for clinical use. However, the sensitivity or specificity of the gendrive assay has not been mentioned, even from pilot studies. 7] Page 7, Lines 16-17: The authors indicate that data collection was between January 2020 and November 2020. How has this study been impacted by the pandemic mitigation efforts? 8] Page 7, Lines 47-19: Please clarify the final sentence in the paragraph. Are your referring to the timing of aminoglycoside
--

	administration or whether an aminoglycoside is actually prescribed pre- and post-implementation? 9] Page 8, Exclusion Criteria: It is my understanding that NICU infants routinely receive immediate gentamicin for at least 48 hours until sepsis or SIRS is ruled out. Those who have a positive culture tend to receive the treatment longer. It would be helpful to describe the standard of care treatment process for the NICUs in this project, and indicate what % of infants will be eligible for study based on treatment protocols and rule-out procedures. 10] Given the new antibiotic stewardships, the use of gentamicin has also reduced dramatically in the NICUs (in US, at least). Do the authors believe that they will still have a robust cohort and meet recruitment goals of this study if less infants are receiving aminoglycoside treatments? is this a concern? 11] Page 9, Testing process and assessment: I have concerns regarding the waiver of consent for this study, given that the outcome of the assay may change the treatment outcome. Are the physicians actually using the test outcome (+ or - for m. 1555A>G) to guide their clinical decisions for prescribing the infant gentamicin? If YES, then I'm confused why the sensitivity of the genetic assay was not mentioned, or why parental consent is not required? This is clearly an experimental protocol, and concerning that it is a proof of concept study already guiding clinical decision making (if Lines 54-59 are correct in their interpretation). [In the discussion section, Page 14, Lines 13-36, the authors appear to address this concern but it is not presented in that light on Page 9. It's also not clear if the physician will know the result of the genetic assay or if they will be blind to outcome. I would present this information earlier so the reader is not misled] 12] Page 10, Sample Size: What % of the 900 would be expected to receive gentamicin (pre-implementation)? 13] Will the parents of the infants eventually be notified of the test outcome? 14] Suggestion to update references to include information about the new antibiotic stewardships in the NICU (e.g., PMC6696821) Thank you for the opportunity to review this wonderful protocol.
--	---

VERSION 1 – AUTHOR RESPONSE

Reviewer 1

This protocol addresses a highly significant problem of genetic susceptibility of neonates to aminoglycoside ototoxicity. It provides an excellent rationale for the study to be done and provides statistical validation of the data to be acquired. The results could later be applied worldwide to prevent hearing loss in neonates with genetic susceptibility to aminoglycoside ototoxicity by providing rapid screening using a novel tool a genetic point of care test. Such a tool could prevent lifelong hearing loss in neonates and promote a substantial improvement in quality of life and reduce health care costs associated with lifelong deafness in infants if alternative antibiotic therapy can be substituted in a timely manner.

Response: No comments to address

Reviewer 2

This protocol reported a prospective observational trial to assess the implementation of the genetic POCT in NICU. The authors state that this is the first attempt to implement a genetic POCT in the acute neonatal setting.

The authors state that when the POCT result is positive, a standard non-gentamicin pathway should be prescribed if antibiotic therapy is required, without detailed description of the non-gentamicin prescription. We doubt this straight-forward determination of antibiotic selection for the infant by genetic POCT, considering that gentamicin is the first-choice antibiotic for neonatal sepsis, and it is not clear whether other antibiotic regimens are equivalent to gentamicin in neonatal sepsis treatment.

Response: In paragraph 4 of the introduction, we now highlight how the UK NICE guidance for early onset neonatal infection recommends a combination of gentamicin and benzylpenicillin as first-line therapy not due to superiority over other agents on an individual patient level, but because that regimen has a narrower spectrum of activity and therefore does not readily contribute towards the development of resistant bacterial pathogens. We also reference a recent article published in the Journal of Medical Ethics where we have recently discussed this issue.

Moreover, there is a conflict between the use of gentamicin that may cause deafness and non-gentamicin prescription that may increase the risk of death, and an opt-out consenting model might be inappropriate.

Response: The PALOH project represents the first time a rapid genetic test has been implemented in the acute neonatal setting to influence management. Therefore, there are several novel ethical and methodological issues raised by the protocol. As discussed above, this very issue has been debated in a recent round table in the Journal of Medical Ethics. We have highlighted this reference and stated the protocol does not create a choice between deafness and increased risk of death. Cephalosporins are equally efficacious, just broader spectrum, therefore not advised as first line agents due to the risk of antibiotic resistance on a population level.

Additionally, this study plans to recruit ~900 babies. Calculating the prevalence of m.1555A>G (1 in 500), it is possible to never detect carriers in this study.

Response: This is correct; however, the study was not designed to explicitly identify an individual with the variant. Rather, the study aims to assess the impact and feasibility of implementing a genetic POCT within clinical practice. As such, no positive patients need to be identified for the study to have value. Of note, 3 positive patients were identified during the PALOH trial [Unpublished].

Finally, the manuscript mentioned: "The trial will start in January 2020 and it is expected that the last participant will be recruited in November 2020". However, protocol papers should report planned or ongoing studies. I am confused that is it a completed study or an ongoing study?

Response: Addressed above in detail.

Reviewer 3

This is a well-written study protocol by a highly esteemed group of scientists. The clinical implementation of a genetic test to detect mutations associated with ototoxicity is important, and may have considerable clinical impact, particularly in the prevention of aminoglycoside-induced ototoxicity. Our current clinical models are moving to point-of-care programs, which tend to be more efficient and cost-effective. This trial is directly in-line with modern practice.

I have reviewed the manuscript in detail and have provided some comments, as well as requests for clarification:

1] Page 5, Line 19: Provide proper citation and reference for NICE guideline (CG149).

Response: Reference Added

2] Page 5, Line 31: Add hyphen "aminoglycoside-induced ototoxicity"

Response: Hyphen Added

3] Page 5, Lines 38-40: Elaborate on comparison between structures of 12S rRNA and 16S rRNA. This statement was made without clear context.

Response: To provide clarity, the sentence beginning on line 40 now reads "This causes a change in the conformation of the 12S rRNA, producing a structure more like the bacterial 16S rRNA, meaning aminoglycosides can more readily bind causing cellular toxicity. The impact of this aberrant binding is most apparent in the inner ear hair cells, leading to ototoxicity".

4] Introduction (general comments): Given the audience of this journal, it would be beneficial to the reader to provide a brief description of both ototoxicity and vestibulotoxicity.

Response: In paragraph 2, the following sentence has been added "An ototoxic compound is one that can cause damage to the auditory system (cochleotoxicity) or, less commonly, vestibular system (vestibulotoxicity)."

5] Page 6, Lines 17-20: If β -lactam antibiotics are equally effective as aminoglycosides, and produce less adverse events, then why do we continue to use the aminoglycoside? This is an important point of discussion and is highly relevant to numerous clinical cohorts routinely treated with aminoglycosides.

Response: As outlined above, in paragraph 4 of the introduction we highlight how the UK NICE guidance for early onset neonatal infection recommends a combination of gentamicin and benzylpenicillin as first-line therapy not due to superiority over other agents on an individual patient level, but because that regimen has a narrower spectrum of activity and therefore does not readily contribute towards the development of resistant bacterial pathogens. We also reference a recent article published in the Journal of Medical Ethics where we have recently discussed this issue.

6] Page 6, Lines 23-30: The authors indicated that the rapid test is comparable to the gold-standard and has been approved for clinical use. However, the sensitivity or specificity of the genedrive assay has not been mentioned, even from pilot studies.

Response: We have removed the phrase "comparable to gold standard" as it is not appropriate to outline the relevant supporting data in this manuscript. The test was awarded CE certification, indicating its appropriateness for clinical use, which hopefully will be sufficient. Data around the analytical validation of the assay will be presented in the final study manuscript.

7] Page 7, Lines 16-17: The authors indicate that data collection was between January 2020 and November 2020. How has this study been impacted by the pandemic mitigation efforts?

Response: The study continued despite the SARS-CoV-2 pandemic, demonstrating the system could be implemented in complex clinical environments, despite a frequently changing landscape. We have indicated this in the methods section (study centres and time schedule).

8] Page 7, Lines 47-19: Please clarify the final sentence in the paragraph. Are you referring to the timing of aminoglycoside administration or whether an aminoglycoside is actually prescribed pre- and post-implementation?

Response: The sentence now reads "In the acute neonatal setting, delivering antibiotics in a timely manner is critical and this should ideally be within an hour after admission. As such, we will assess whether the time to antibiotic administration after admission is impacted by implementation of the genetic POCT. A reference time to antibiotic was measured over a month-long period before implementation."

9] Page 8, Exclusion Criteria: It is my understanding that NICU infants routinely receive immediate gentamicin for at least 48 hours until sepsis or SIRS is ruled out. Those who have a positive culture tend to receive the treatment longer. It would be helpful to describe the standard of care treatment process for the NICUs in this project, and indicate what % of infants will be eligible for study based on treatment protocols and rule-out procedures.

Response: In the Study Centres section, we have now highlighted that "Both units use a combination of a beta-lactam and an aminoglycoside as first line therapy for early onset neonatal infection". In the inclusion criteria, we highlight that all babies admitted to NICU across the two participating sites, commencing from the trial start date, will be eligible for recruitment. Based on the limited exclusion criteria, we expect the majority of admissions to be eligible for recruitment, which we have emphasised in the exclusion criteria section. We are unable to provide an exact proportion, however these criteria were derived in consultation with neonatologists to acknowledge those rare cases where clinicians could, and would want to, deliver antibiotics immediately on admission.

10] Given the new antibiotic stewardships, the use of gentamicin has also reduced dramatically in the NICUs (in US, at least). Do the authors believe that they will still have a robust cohort and meet recruitment goals of this study if less infants are receiving aminoglycoside treatments? is this a concern?

Response: This is an important consideration, but not a great concern due to the widespread use of gentamicin in the UK. Based on the study cohort, we foresee no issues with recruitment. However, we acknowledge geographical variation will impact the utility and relevance of this particular assay. In the study centres section we acknowledge that "antibiotic choice and frequency of administration vary between units, nationally and internationally".

11] Page 9, Testing process and assessment: I have concerns regarding the waiver of consent for this study, given that the outcome of the assay may change the treatment outcome. Are the physicians actually using the test outcome (+ or - for m. 1555A>G) to guide their clinical decisions for prescribing the infant gentamicin? If YES, then I'm confused why the sensitivity of the genetic assay was not mentioned, or why parental consent is not required? This is clearly an experimental protocol and concerning that it is a proof of concept study already guiding clinical decision making (if Lines 54-59 are correct in their interpretation).

[In the discussion section, Page 14, Lines 13-36, the authors appear to address this concern but it is not presented in that light on Page 9. It's also not clear if the physician will know the result of the genetic

assay or if they will be blind to outcome. I would present this information earlier so the reader is not misled]

Response: This is an interesting point which we have considered extensively over the past few years. For clarity, clinicians have used this genetic test result to inform clinical decision making regarding antibiotic choice. The clinicians are not blind to the outcome. The reviewer's initial concerns are entirely understandable. This represents the first time that a genetic point of care test has been used to alter management in the acute neonatal setting. The system itself was CE certified for clinical use, and therefore not an experimental system. Furthermore, testing for the m.1555A>G is common in certain areas of clinical practice, the technology has simply never existed before now to test this rapidly.

Given the novel nature of this protocol, we recently wrote an editorial round table for the Journal of Medical Ethics, which is now cited in the ethics section. This describes, in detail, the ethico-legal approach to this study. Responding to the reviewer's point about presenting the concept earlier, we now highlight on Page 10 that "Once the assay result is available, the result can be used to then undertake personalized antibiotic prescribing, avoiding aminoglycoside antibiotics if the m.1555A>G variant is detected. The CE certified assay will be integrated into the admission pathways at both centres and clinicians will not be blinded to the results."

12] Page 10, Sample Size: What % of the 900 would be expected to receive gentamicin (pre-implementation)?

Response: In the Study Centers and Time Schedule Section, we now write "Based on historical admission data, the majority of admissions to both units will be screened for infection and, as such, considered for antibiotic therapy"

An exact % figure is not available pre-implementation at each site, however the most recent National Neonatal Audit Programme found that 104,577 babies were admitted to Neonatal Intensive Care Units (NICUs) in 2018. A large proportion of these patients will be investigated and treated for early onset neonatal infection, approximately 10% of all newborn babies.

It should be noted that there is considerable variation in antimicrobial prescribing practices both between and within different countries. The type of antibiotics used, dosing regimens, and monitoring protocols can vary between different nations, cities, and institutions. The reasons for this are multifactorial, associated with historical practice, the adoption of clinical guidelines, and economic considerations. Variability in the consumption of AGs may also be related to broader, non-prescription related, availability.

13] Will the parents of the infants eventually be notified of the test outcome?

Response: Any positive result will trigger a referral to the clinical genetics service, as outlined in paragraph 3 of the testing process and assessment section. For clarity, in the same paragraph, we now confirm that "Negative results will not be formally reported to the parents. However, all parents will be provided with information on the study in their NICU information pack, allowing them to contact the study team for further information if they wish."

14] Suggestion to update references to include information about the new antibiotic stewardships in the NICU (e.g., PMC6696821)

Response: Many thanks. This reference has now been cited in the Study Centres section, highlighting antibiotic stewardship and variation in practice.

VERSION 2 – REVIEW

REVIEWER	Peng, Zhiyu Beijing Genomics Institute Shenzhen
REVIEW RETURNED	17-Mar-2021

GENERAL COMMENTS	I still concern about the design of this study, particularly the number of recruited babies. Although the authors described one of the limitations is that “the variant is never detected throughout the study, meaning some secondary outcomes cannot be reliably assessed.” The manuscript presents the primary objective of this study is “to critically assess the performance of a newly developed genetic POCT for use on the NICU and measure whether the clinical teams can integrate the test into their clinical practice without disrupting normal standard of care, facilitating tailored antibiotic prescribing.” As stated by the authors, the prevalence of this variant is 1 in 500. Recruiting 900 babies, the maximum number of babies with the target variant is two. My concern is that both the primary and secondary outcome of this study is hard to evaluate.
--

REVIEWER	Garinis, Angela Oregon Health & Science University
REVIEW RETURNED	22-Mar-2021

GENERAL COMMENTS	The authors did an excellent job addressing the reviewers comments, and the additional text has improved the manuscript. The use of point-of-care options, particularly for genetic markers, is timely to reduce mitigate toxicities and optimize clinical outcomes. I only have one further suggestions to improve the rationale for this project: 1] The authors briefly discuss that prescribing practices of aminoglycosides may differ across different regions of the world . Although this is true, more justification is requested to elucidate this point. Antibiotic stewardships are changing the optimal selection, dosage, and duration of aminoglycoside use in NICU babies. The use of these treatments have declined in the NICU, particularly in the US, to reduce the potential toxicities associated with treatment. It would be helpful if the authors provided a brief description and rationale for investing in the genetic POCT test for m.1555A>G variant given this potential reduction in aminoglycoside use across NICUs (cost - benefit description)
--

VERSION 2 – AUTHOR RESPONSE

Reviewer 2

“I still concern about the design of this study, particularly the number of recruited babies. Although the authors described one of the limitations is that “the variant is never detected throughout the study, meaning some secondary outcomes cannot be reliably assessed.”

‘The manuscript presents the primary objective of this study is “to critically assess the performance of a newly developed genetic POCT for use on the NICU and measure whether the clinical teams can

integrate the test into their clinical practice without disrupting normal standard of care, facilitating tailored antibiotic prescribing'

As stated by the authors, the prevalence of this variant is 1 in 500. Recruiting 900 babies, the maximum number of babies with the target variant is two. My concern is that both the primary and secondary outcome of this study is hard to evaluate.

Response: We appreciate the reviewer's request for further clarification. However, no trial outcome detailed within the manuscript requires the detection of a positive genotype.

To provide more detail, as stated at the previous review stage, the PALOH trial was not designed to explicitly identify an individual with the m.1555A>G variant. Rather, the trial is designed to examine whether a rapid novel genetic point of care test can be implemented into the acute setting, without disrupting normal standard of care. The analysis is still entirely possible, and equally as valid, even if a m.1555A>G genotype is not detected throughout the entirety of the study. The identification of a negative result (i.e. m.1555A) will still inform tailored antibiotic prescribing (i.e. there will be increased confidence that aminoglycosides are safe) and measuring the way in which this information is used in a clinical setting is important.

In summary, the outcomes of this study do not require the identification of the m.1555A>G genotype. As the reviewer highlights, the prevalence of the variant is 1 in 500, meaning any trial design which required identification of m.1555A>G would require an infeasibly high sample size. Accordingly, the PALOH study has been designed to measure the impact of implementing a genetic technology in the acute setting.

For clarity, we have outlined each outcome below, detailing whether detection of the m.1555A>G variant is relevant to evaluation of that outcome.

- Primary Outcome: The primary outcome is the number of neonates who are successfully tested for the m.1555A>G variant via this novel POCT genetic test as a proportion of all babies who receive aminoglycoside antibiotics on the two participating NICUs. – This outcome includes all babies tested and therefore detection of the m.1555A>G variant is not relevant.
- Secondary Outcome 1: This trial will also assess several secondary outcomes including the total number of neonates identified with the m.1555A>G genetic variant (positive result) as a proportion of the study population. – Although this outcome refers to the detection of positive variants, it is simply a quantitative measure. If no positive genotypes were detected, then this would be reported and the rest of the study would not be impacted.
- Secondary Outcome 2: An assessment of resource impact will be undertaken, including a health economic analysis of additional staff time to secure sample and testing and costs associated with alternative antibiotic prescribing for any positive tests. – This is a health economic assessment therefore the results will be reviewed as they occur. The detection of a positive genotype is not required for an assessment to take place.

- Secondary Outcome 3: We will record the number of neonates where testing was not undertaken and the rationale for this. – A positive genotype is not required to adequately assess this outcome.
- Secondary Outcome 4: Correlation of the POCT result with the current in-house reference assay will be assessed and any test fails will be reported – This is a laboratory assay confirming the genotypes detected by the POCT, therefore a positive genotype is not required.
- Secondary Outcome 5: We will assess the whether the variance in timings of time to antibiotic administration after admission is impacted by implementation of the genetic POCT – This is a measure of clinical timings and therefore the genotype result is not relevant.

In summary, no trial outcome detailed within the manuscript requires the detection of a positive genotype. The sample size of 900 is entirely sufficient to adequately study the outcomes listed above.

Reviewer 3

The authors did an excellent job addressing the reviewers' comments, and the additional text has improved the manuscript. The use of point-of-care options, particularly for genetic markers, is timely to reduce mitigate toxicities and optimize clinical outcomes. I only have one further suggestion to improve the rationale for this project:

1] The authors briefly discuss that prescribing practices of aminoglycosides may differ across different regions of the world. Although this is true, more justification is requested to elucidate this point. Antibiotic stewardships are changing the optimal selection, dosage, and duration of aminoglycoside use in NICU babies. The use of these treatments have declined in the NICU, particularly in the US, to reduce the potential toxicities associated with treatment. It would be helpful if the authors provided a brief description and rationale for investing in the genetic POCT test for m.1555A>G variant given this potential reduction in aminoglycoside use across NICUs (cost - benefit description).

Response: Many thanks for this comment. In response, we have dedicated a paragraph in the discussion reviewing this topic. It is beyond the scope of this work to provide a formal economic justification, but we agree that the concept warrants discussion. The new text in the discussion now reads;

“It should be noted that there is considerable variation in antimicrobial prescribing practices both between and within different countries. The type of antibiotics used, dosing regimens and monitoring protocols can vary between different nations, cities, and institutions. The reasons for this are multifactorial, associated with historical practice, the adoption of clinical guidelines, and economic considerations. Where there are national guidelines, such as those produced by NICE, one would expect relative consistency in prescribing practice. However, where practice is less guideline driven, there may be more variation. This is exemplified by a retrospective cohort study of 52,061 infants from 127 NICUs across California, which found that antibiotic use varied between centers by up to 40-fold. Variability in prescribing practice in the UK is likely to be less pronounced, given the more guideline driven, nationalised nature of its healthcare system. As such, health economic cases for the utilization of POCT systems, such as the one described here, will vary depending on the wider healthcare context and should be formally examined locally.”

Once again, many thanks for your thorough review of this manuscript. We look forward to receiving your response.